# REM-Enriched Naps Are Associated with Memory Consolidation for Sad Stories and Enhance Mood-Related Reactivity

**DOI:** 10.3390/brainsci6010001

**Published:** 2015-12-29

**Authors:** Médhi Gilson, Gaétane Deliens, Rachel Leproult, Alice Bodart, Antoine Nonclercq, Rudy Ercek, Philippe Peigneux

**Affiliations:** 1UR2NF—Neuropsychology and Functional Neuroimaging Research Unit, avenue F.D. Roosevelt 50, Bruxelles 1050, Belgium; gaetane.deliens@ulb.ac.be (G.D.); rleproul@ulb.ac.be (R.L.); alice.bodart@ulb.ac.be (A.B.); 2UNI—ULB Neuroscience Institute, Université Libre de Bruxelles (ULB), avenue F.D. Roosevelt 50, Bruxelles, Belgium; 3CO3—Consciousness, Cognition & Computation Group, avenue F.D. Roosevelt 50, Bruxelles 1050, Belgium; 4LISA—Laboratories of Image, Signal processing and Acoustics, avenue F.D. Roosevelt 50, Bruxelles 1050, Belgium; anoncler@ulb.ac.be (A.N.); rercek@ulb.ac.be (R.E.)

**Keywords:** sleep, memory, emotion, affect, rapid eye movement, sleep spindles, electrodermal activity

## Abstract

Emerging evidence suggests that emotion and affect modulate the relation between sleep and cognition. In the present study, we investigated the role of rapid-eye movement (REM) sleep in mood regulation and memory consolidation for sad stories. In a counterbalanced design, participants (*n* = 24) listened to either a neutral or a sad story during two sessions, spaced one week apart. After listening to the story, half of the participants had a short (45 min) morning nap. The other half had a long (90 min) morning nap, richer in REM and N2 sleep. Story recall, mood evolution and changes in emotional response to the re-exposure to the story were assessed after the nap. Although recall performance was similar for sad and neutral stories irrespective of nap duration, sleep measures were correlated with recall performance in the sad story condition only. After the long nap, REM sleep density positively correlated with retrieval performance, while re-exposure to the sad story led to diminished mood and increased skin conductance levels. Our results suggest that REM sleep may not only be associated with the consolidation of intrinsically sad material, but also enhances mood reactivity, at least on the short term.

## 1. Introduction

Memory consolidation is the process by which newly learned and fragile information is progressively converted into more robust, steadier representations for long-term memory storage [1]. Importantly, emotional processes modulate memory consolidation as well as encoding and retrieval [2,3]. As a result, arousing emotionally loaded memories are often better remembered than neutral memories that lack a specific affective component [4]. Although it is now widely recognized that post-learning sleep contributes to memory consolidation [5,6], the sleep-dependent mechanisms underlying emotional regulation and consolidation of emotionally tinted memories represent a more recent but growing domain of interest [7,8,9].

Several studies suggest that sleep is important for affective regulation. For instance, poor sleep quality results in decreased cognitive reappraisal [10,11,12] and enhanced reactivity toward aversive stimuli. Accordingly, larger pupillary responses to negative pictures are observed after sleep deprivation [13]. In addition, amygdala activity and its connectivity with brainstem regions associated with autonomic activation are enhanced after sleep deprivation, whereas connectivity with the medial prefrontal cortex deteriorates [14]. This suggests that sleep deprivation impairs the top-down inhibitory control of emotions, eventually exacerbating autonomic reactivity.

Specific associations have been proposed between rapid-eye movement (REM) sleep and the regulation of arousing emotions. Indeed, selective REM sleep deprivation for one night is associated with an increased likelihood to perceive negative pictures from the International Affective Picture System (IAPS, [15]) as threatening, whereas emotional reactivity remains stable after non-REM (NREM) sleep interruptions [16]. In the latter study [16], functional magnetic resonance imaging data disclosed a global decrease in brain reactivity in occipital and temporal regions involved in emotion processing after NREM interruptions, whereas brain activity remained stable or even increased after REM sleep deprivation. The depotentiating effect of sleep on emotional reactivity in response to picture viewing is also corroborated by skin conductance measures [17]. Re-exposure to neutral and negative pictures after a 12-h sleep interval results in decreased electrodermal responses for both picture types, while no effect is observed after a wake interval. Contrarily however, enhanced emotional reactivity and increased negative valence ratings for emotional pictures were reported after late-night sleep periods rich in REM sleep [18]. Likewise, participants with longer late night REM sleep exhibited higher emotional post-sleep reactivity to an aversive film, as assessed by skin conductance and electromyographic measurements [19]. Therefore, and although effects are mitigated, available data suggest that REM sleep participates in emotional processing and its neural substrates. Other studies rather suggest a role for NREM sleep in the regulation of emotions. Overnight emotional attenuation following the viewing of a stressful film is correlated with increased slow wave sleep (SWS) acquired during the night [20]. Following a daytime nap, NREM sleep occurrence was associated with greater electromyographic habituation to negative stimuli whereas REM sleep occurrence was associated with diminished skin conductance habituation, suggesting that sleep promotes emotional adjustment at the level of somatic responses [21]. In a partial REM sleep deprivation paradigm, subjects exhibiting the highest decrease in the percentage of REM sleep were paradoxically those showing the best adaptation to negative pictures on arousal ratings [22]. Hence, sleep might play a role in the regulation of arousing emotions, but the specific contributions of REM and NREM sleep stages remain to be ascertained.

Along with its regulatory role in emotional processes, sleep contributes to declarative memory consolidation [6]. Furthermore, several studies suggest that post-learning sleep is more beneficial for the consolidation of emotional than neutral memories [23,24], even when recall is tested up to four years after learning [25]. Nevertheless, memory for aversive pictures is usually more resistant to the effects of sleep deprivation than memory for neutral images [26]. Accordingly, sleep deprivation was found to impair free recall for positive movie clips, whereas memory for neutral and negative movie clips was unaltered [27]. It was proposed that a better resistance of negative, potentially dangerous, memories to the effects of sleep deprivation is due to the differential involvement of a complementary amygdalo-cortical network that keeps track of emotional information [28]. Regarding the contribution of specific sleep stages, there is evidence that REM sleep plays a role in the consolidation of emotional memories. For instance, emotionally loaded texts are better remembered than neutral ones after a sleep episode during the second half of the night (*i.e*., late sleep), which is richer in REM sleep [29]. In addition, the consolidation of aversive pictures benefits more from a post-training nap than neutral pictures; an effect correlated with both the percentage of REM sleep and the power of right prefrontal theta oscillations during REM sleep [30]. Aside from a reduction in SWS and a concomitant impairment in the hippocampal-dependent consolidation of neutral texts, the administration of the cortisol synthesis inhibitor metyrapone during the second part of the night (rich in REM sleep) enhances the amygdala-dependent formation of emotional memories [31]. It suggests that cortisol, a stress hormone, may protect emotional memories against an exaggerated emotional surge during REM sleep-related memory consolidation processes. At variance however, other studies show that late-night REM sleep limits the decrease in negative valence ratings, even if sleep [32] and REM sleep [33] indeed contribute to the consolidation of emotional memories.

Because recent studies highlight an association between REM sleep duration and an attenuated decrease in emotional reactivity, it is tempting to suggest that sleep, and REM sleep in particular, plays a protective role for the valence of a stimulus. This role might be possibly paralleled by a supporting role of sleep in the consolidation of the associated memory. Surprisingly however, the potential effect of sleep after having experienced less specifically arousing emotional events, such as sad stories, has not been experimentally investigated. Nonetheless, clinical studies suggest altered NREM and REM sleep in mood altered psychiatric conditions [34], and REM sleep and dreaming seem to moderate the overnight changes in mood states in healthy volunteers [35].

In this framework, the present study aimed at further investigating the contribution of sleep—and especially REM sleep—to the consolidation of neutral and sad stories and the regulation of associated mood reactivity. To do so, participants took either a short or long nap in the morning after a moderate sleep restriction on the night preceding the testing, a procedure known to favor the occurrence of REM sleep [36,37]. Across two counterbalanced sessions, each participant learned either a neutral or a sad story, followed by the short or long nap condition under polysomnographic recording. Memory for the story was then tested after the nap. Additionally, mood and emotional reactivity to the story were tested before and after the nap. Since lack of differentiation between the evaluation of mood states and the affective evaluation of the content to remember is a potential confounding factor, we specifically investigated (a) how participants judged the emotional content of the story before and after the nap; (b) how their own subjective mood and arousal states were influenced by the nap; and (c) how objective reactivity—as measured by electrodermal activity—was influenced by the nap.

## 2. Experimental Section

### 2.1. Participants

Twenty-seven healthy participants gave their written informed consent to participate in this study approved by the Faculty ethics committee at the Université Libre de Bruxelles. Three participants were excluded from the study due to insufficient sleep during the naps (<10 min). The 24 remaining participants (5 males), age (mean ± SD) 21.8 ± 1.8 years were native French speakers, non-smokers, free of medications likely to influence mood and/or sleep quality and without reported history of neurological or psychiatric disorders. A pre-study examination disclosed good to moderate habitual sleep quality (Pittsburgh Sleep Quality Index ≤ 8, [38]), no excessive daytime sleepiness (Epworth Sleepiness Scale ≤ 11, [39]), no severe mood disorder (Beck’s Depression Scale ≤ 12, [40]; Anxiety ≤ 12 and Depression ≤ 12 Scales, [41]), verbal intelligence within normative values (range within 25–40, Mill-Hill Vocabulary Scale, [42]), a mean alexithymia score of 45.58 ± 8.6 (no participant above the cut-off of 61; Alexithymia Toronto Scale, [43]) and a mean chronotype score of 45.81 ± 9.1 (one extreme evening type participant; Morningness-Eveningness Questionnaire range 27–59, [44]). No difference between participants in the long and short nap conditions was evidenced for any of these values (all *p* > 0.1). Participants were instructed to follow regular sleep schedules before and throughout the experiment, and compliance was verified using wrist actimetry (Daqtometer, Daqtix GbR, Oetzen, Germany) and daily sleep logs. The night before each of the two testing sessions, they were asked to restrict their total sleep time by 2 h to favor the appearance of REM sleep during the morning naps. Participants were also prevented from drinking caffeinated, stimulant or alcoholic drinks during the entire study.

### 2.2. Procedure

Participants were assigned to either the short (*N* = 12) or long (*N* = 12) nap condition. Each of them completed two randomized sessions, one to learn the neutral story and the other to learn the sad story. The two sessions were separated by at least one week. A typical session is summarized in Figure 1. One week prior to the session, participants received an actimeter and were given instructions to keep regular sleep schedules, follow their usual bedtimes, and wake up 2 h earlier than their usual wake time on the day of the testing session. On the session day, participants arrived at the sleep laboratory at 8:00 a.m. After being prepared for polysomnographic (PSG) recording, they were asked to wash their hands to obtain an optimal skin conductance signal. At 8:30 a.m., subjective sleepiness and objective vigilance were assessed using the Karolinska Sleepiness Scale (KSS, [45]) and the 10-min version of the Psychomotor Vigilance Task (PVT, [46]), respectively. They then completed the Self-Assessment Manikin (SAM, [47]) for mood and arousal. At 9:00 a.m., skin conductance (SC) was recorded for 5 min at rest (baseline). The story was then delivered over loudspeakers for 10 min while SC continued to be recorded. Participants were instructed to carefully listen to the story and to let themselves be emotionally engaged. They were also informed that memory for the story would be tested after the nap. After listening to the story, participants filled out the SAM questionnaire (SAM 2) again. They also had to assess on a seven-point Likert scale from 1 (“not at all”) to 7 (“absolutely”) the 6 basic emotions (anger, surprise, disgust, fear, joy, and sadness) as identified by Ekman, Friesen and Ellsworth [48] to describe the stories (Rating 1). They were also asked to rate the emotional valence (positive or negative) of the stories on a Visual Analogous Scale (VAS 1). They were then requested to take either a short or long nap. Participants were awakened after approximately 45 or 90 min of sleep, depending on the nap condition they were assigned. They were awakened in this period when the experimenter noticed an arousal or a micro arousal on the PSG recording. The PSG setup was then removed and participants had the opportunity to wash their hair. At 11:20 a.m. (approximately 40 or 85 min after wake-up time, according to nap duration), sleepiness (KSS 2), vigilance (PVT 2) and mood/arousal (SAM 3) were again assessed, followed by the 25-item questionnaire to assess the recall of the story. Participants then listened to the story for the second time under SC measurement to assess emotional reactivity, and SAM 4 was completed as well as Rating 2 and VAS 2. One week later, they returned to the lab and underwent the same session but with the other story type.

**Figure 1 brainsci-06-00001-f001:**
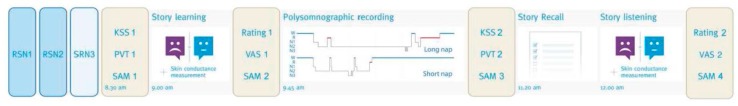
Time course for one session. RSN = Regular Sleep Night (8 h); SRN = Sleep Restriction Night (6 h); KSS = Karolinska Sleepiness Scale; PVT = Psychomotor Vigilance Task; SAM = Self-Assessment Manikin; Rating = rating of the 6 emotions; VAS = Visual Analogous Scale.

### 2.3. Material

#### 2.3.1. Stories and Questionnaires

The audio stories were selected from the Brussels Mood Inductive Audio Stories (BMIAS) database [49]. The neutral story described a giraffe and its natural environment, a story with no significant influence on mood or arousal states. The sad story related the terminal stage of a young woman suffering from cancer, a story that significantly modulates mood state, as reflected by increased negative mood scores [49]. Both stories lasted 10 min. To test memory for these stories, two 25-item questionnaires were created, with questions about the context (e.g., How was the weather?) or specific elements (e.g., who administered the injection?).

We conducted an independent pre-test to check that the two stories presented the same learning difficulty levels. The pre-test was run with 30 participants (28 women, 2 men; mean age 21.5 ± 1.7 years). Half of them (*N* = 15) listened to the neutral story, and the other half (*N* = 15) listened to the sad story. They were instructed to memorize it and let themselves be emotionally engaged in the story. One hour later, they were asked to respond to the 25-item questionnaire in a recall session. Memory performance was similar for both stories (neutral: 61.86% ± 12%, sad: 59.72% ± 14.6%, independent-sample t-test: *t*(28) = 0.43; *p* = 0.665), suggesting equal difficulty.

#### 2.3.2. Skin Conductance Measurement

Skin conductance was recorded using a lab-made monitoring device designed specifically by two signal acquisition experts (Antoine Nonclercq, Rudy Ercek) according to recent recommendations for electrodermal measurements [50]. The device monitors DC conductance using a Wheatstone bridge, on which 500 mV is applied. The signal is low-pass filtered at 1.6 Hz and sampled at 10 Hz with 10 bits resolution. A digital low-pass filter (moving average) is then applied. The system was calibrated before each recording session. Linear regression, obtained from a set of standard resistors chosen in the skin conductance range (from 100 kΩ to 1 MΩ; *i.e*., from 10 µS to 1 µS), was used for calibration. In each session, electrodermal activity was first recorded during a 5 min rest period, during which participants were asked to stay calm and relax (baseline). Immediately afterward, the story was presented for 10 min. Electrodermal recordings were split into four equal portions (P1 from 0 to 150 s, P2 from 150 to 300 s, P3 from 300 to 450 s, P4 from 450 to 600 s), each expressed relative to the 5-min baseline period (e.g., (skin conductance levels (SCL)—Baseline SCL)/Baseline SCL).

#### 2.3.3. Polysomnography

Participants were prepared for polysomnographic recording (BioSemi B. V., Amsterdam) upon their arrival to the sleep laboratory (8.00 a.m.). The recording included four electroencephalographic (EEG) derivations (Fp1-M2, Fp2-M1, C3-M2 and O1-M2), two electrooculograms (EOGs) and two chin electromyograms (EMGs). Data were recorded at a sampling rate of 2048 Hz. High-pass frequency filters were set at 0.30 Hz and low-pass frequency filters were set at 45 Hz for EEG and EOG recordings. EMG high-pass and low-pass frequency filters were respectively set at 10 and 100 Hz. The nap took place in an insulated, quiet room. Each 30-second epoch of recording was scored as Wake stage, NREM stage 1 (N1), N2, N3 or REM sleep stage following standard criteria [51] using the PRANA software (PhiTools, Strasbourg, France) by an experienced technician (MG) who was not aware of the content of the story listened to before the nap. Sleep period time (SPT) was defined as the time elapsed between lights off to lights on, total sleep time (TST) was defined as the time elapsed between the first and the last epochs scored as sleep (N1, N2, N3 or REM), wake after sleep onset (WASO) was defined as the amount of wake time elapsed between sleep onset and sleep offset, sleep onset latency (SOL) was defined as the time elapsed between lights off and the first epoch of N1, and sleep efficiency was defined as the TST expressed in percent of the SPT.

### 2.4. Data Analysis

#### 2.4.1. Actigraphic Data

Total sleep time (TST) was estimated for each night via wrist actimetry recordings. Sleep onset was determined when a significant decrease in ambient light, suggesting lights off, was followed by a decrease in movement (90% reduction of the count of movement by minute, in comparison to the preceding 15 min) for a minimum of 15 min. Sleep offset was detected when a significant increase in motor activity (90% increase of the count of movement by minute) was observed for a minimum of 15 min.

#### 2.4.2. Sleep Data

Sleep spindles and rapid eye movements (REMs) were detected through automated PRANA algorithms, then visually verified and scored by a trained technician (MG). Sleep spindles were identified for each subject on channel C3 during N2. The EEG signal was band-pass filtered with a 5th order IIR filter between 11 and 13 Hz for slow spindles and between 13 and 16 Hz for fast spindles. Sleep spindles comprised of a duration between 500 and 3000 ms were then detected. We calculated the total count and the density (spindles count per minute) for slow and fast sleep spindles. REMs were detected during REM sleep as a strong (>25 microvolts), short (<1000 ms) downward and convergent movements of both eyes. Additionally, spectral absolute power analysis was computed in the following frequency bands: Delta (0.75–4.5 Hz), Theta (4.5–8.5 Hz), Alpha (8.5–12.5 Hz), Sigma (12.5–15.5 Hz), Beta (15.5–22.5 Hz) and Gamma (22.5–45.5 Hz). Spectral power analyses were first computed for N2, N3 and REM sleep separately then for NREM sleep (N2 + N3) and finally for all sleep stages (N2 + N3 + REM). All sleep analyses were performed after automated and visual artifact removal.

#### 2.4.3. Statistical Analyses

Statistical analyses were performed using Statistica 7.0 (Statsoft Inc., Tulsa, OK, USA). We conducted repeated and mixed measures analysis of variance tests (ANOVAs), followed by Tukey’s *post-hoc* tests. Significance was set at *p* ≤ 0.05, two-tailed.

## 3. Results

### 3.1. Sleep and Vigilance

#### 3.1.1. Nocturnal Sleep Prior to Experimental Sessions

Subjective total sleep time (TST), as reported in sleep journals, was assessed for each of the three nights preceding both testing sessions (Table 1). A repeated measures ANOVA on subjective TST with Night (nights 1–3) and Story (Neutral *vs.* Sad story) as within-subject factors and Group (Short *vs.* Long nap) as the between-subject factor disclosed a significant effect of Night (*F*(5, 36) = 31.54, *p* < 001). Tukey’s *post-hoc* disclosed a shorter TST during the sleep restriction night (night 3), as compared to the two regular nights (nights 1 and 2, all *p* < 0.001) without any significant difference between night 1 and night 2 (*p* > 0.9). The main effects of Group (*F*(1, 18) = 1.05, *p* = 0.320) and Story (*F*(1, 18) = 0.905, *p* = 0.458) were non-significant. We observed a significant interaction between Group and Story (*F*(1, 18) = 4.58, *p* = 0.046), but *post-hoc* tests failed to reach significance (all *p* > 0.1). In agreement with subjective TST data, a repeated measures ANOVA conducted on TST derived from actimetric data (Table 1) revealed no significant effect for Group (*F*(1, 19) = 0.97, *p* = 0.338) or Story (*F*(1, 19) = 0.96, *p* = 0.339), but a significant effect of Night (*F*(2, 38) = 21.87, *p* = 0.000). Tukey’s *post-**hoc* disclosed a shorter TST during the sleep restriction night, as compared to the two regular nights (all *p* < 0.001). No differences were observed between night 1 and night 2 (*p* > 0.9) and all interactions were non-significant (all *p* > 0.1).

**Table 1 brainsci-06-00001-t001:** Subjective and objective Total Sleep Time (mean ± SD, hours), derived from sleep agendas and actimetry, respectively. Data were obtained during the three nights preceding the testing sessions.

	Group	Story	Night 1 (RSN)	Night 2 (RSN)	Night 3 (SRN)
TST (agenda, in hours)	Short nap	Neutral story	8.21 ± 1.2	8.08 ± 1	5.83 ± 1 *
		Sad story	7.67 ± 1.7	7.75 ± 1.7	5.58 ± 1.2 *
	Long nap	Neutral story	7.89 ± 1.1	7.25 ± 0.8	5 ± 0.9 *
		Sad story	7.5 ± 1.5	7.89 ± 1.7	5.78 ± 1 *
TST (actimetry, in hours)	Short nap	Neutral story	8.58 ± 0.8	8.47 ± 1.1	6.26 ± 0.7 *
		Sad story	8.82 ± 2.9	8.69 ± 2	6.11 ± 1 *
	Long nap	Neutral story	8.15 ± 2.1	7.87 ± 1	6.14 ± 2.3 *
		Sad story	8.31 ± 1.7	8.33 ± 1.4	6.47 ± 1.7 *

RSN = Regular Sleep Night; SRN = Sleep Restriction Night; TST = Total Sleep Time; * significant difference with N1 and N2, *p* < 0.05.

#### 3.1.2. Sleep during Long and Short Naps

Sleep measures during the Short and Long nap conditions are separately reported in Table 2 for both the Neutral and Sad story conditions. Repeated measures ANOVAs were computed for all sleep measures with the within-subject factor Story (Neutral *vs.* Sad story) and the between-subject factor Group (Short *vs.* Long nap).

**Table 2 brainsci-06-00001-t002:** Diurnal sleep (naps) macro-and micro-structural measures (mean (SD).

	Long Nap Condition (*N* = 12)		Short Nap Condition (*N* = 12)		Nap	Story	Interaction
	Sad Story	Neutral Story	Sad Story	Neutral Story	*p* Value	*p* Value	*p* Value
W (min)	24.41 (± 11.9)	21.25 (± 15.2)	30.29 (± 35.6)	27.25 (± 24.7)	0.496	0.498	0.989
N1 (min)	11 (± 6.1)	11.71 (± 5.9)	5.5 (± 4)	7.66 (± 4.3)	**0.008**	0.290	0.588
N2 (min)	48.96 (± 16.2)	51.83 ± 14.5)	29.25 (± 11.6)	34.5 (± 9.2)	**0.001**	0.156	0.672
N3 (min)	11.58 (± 15)	6.21 (± 8.4)	7.92 (± 9.5)	1.66 (± 2.7)	0.225	**0.022**	0.855
REM (min)	17.67 (± 9.3)	21.33 (± 11.3)	3.42 (± 7.2)	0.87 (± 1.6)	**0.000**	0.815	0.204
SPT (min)	113.62 (± 12.6)	112.33 (± 8)	76.375 (± 30.9)	71.96 (± 22.4)	**0.000**	0.524	0.726
TST (min)	89.21 (± 11)	91.08 (± 11.9)	46.08 (± 15.3)	44.71 (± 10.2)	**0.000**	0.935	0.578
SOL (min)	13.5 (± 8.9)	9.17 (± 5.1)	9.42 (± 11.3)	8 (± 6.4)	0.404	(0.052)	0.308
WASO (min)	10.92 (± 9.5)	12.125 (± 10.74)	20.87 (± 30.6)	16.27 (± 17.7)	0.365	0.922	0.726
Micro arousals	3.41 (± 0.9)	3.25 (± 0.7)	2.17 (± 0.9)	1.92 (± 0.7)	0.175	0.719	0.942
SE (%)	78.88 (± 9.6)	81.54 (± 12.1)	67.82 (± 26.2)	66.56 (± 21)	(0.07)	0.827	0.544
Total Nr of REMs	139.25 (± 114.7)	142.67 (± 74.4)	30 (± 59.9)	9.08 (± 14.6)	**0.000**	0.684	0.572
REM sleep density	6.83 (± 3.8)	8.27 (± 6.2)	2.56 (± 4.4)	2.94 (± 3.9)	**0.005**	0.433	0.645
Total Nr of Slow Spindles	47.5 (± 21.6)	59 (± 40.4)	17.67 (± 15.9)	20.92 (± 18.4)	**0.001**	0.213	0.480
Total Nr of Fast Spindles	77.83 (± 52.5)	99.42 (± 68.8)	38.58 (± 44.8)	55.42 (± 68.4)	(0.061)	0.125	0.845
Total Nr of Slow & Fast spindles	125.33 (± 67.4)	158.42 (± 100.2)	57.08 (± 56.3)	76.42 (± 81.4)	**0.012**	0.125	0.679
Density of slow spindles	1.09 (± 0.6)	1.28 (± 1.3)	0.58 (± 0.4)	0.61 (± 0.5)	**0.019**	0.633	0.735
Density of fast spindles	1.82 (± 2.4)	1.99 (± 1.6)	1.32 (± 1.1)	1.44 (± 1.3)	(0.096)	0.194	0.431
Density of slow and fast spindles	3.11 (± 2.9)	3.27 (± 2.8)	1.72 (± 1.2)	2.06 (± 1.6)	(0.051)	0.714	0.895

W = Wake; N1 = sleep stage NREM1; N2 = sleep stage NREM2; N3 = sleep stage NREM3; REM = Rapid Eye Movement sleep; SPT = Sleep Period Time; TST = Total Sleep Time; SOL = Sleep Onset Latency; WASO = Wake After Sleep Onset; SE = Sleep Efficiency. Trends < 0.1 are marked between brackets, *p*-values in bold font are significant after correction for multiple comparisons.

At the macrostructural level, a main Group effect was observed for TST (Long nap 90.15 ± 11.3 *vs.* Short nap 45.40 ± 12.8 min, *F*(1, 22) = 118.73, *p* < 0.001), REM sleep duration (Long nap 19.5 ± 10.1 min *vs.* Short nap 2.15 ± 5.2 min, *F*(1, 22) = 54.268, *p* < 0.001), N2 duration (Long nap 50.40 ± 15.2 min *vs.* Short nap 31.87 ± 10.6 min, *F*(1, 22) = 16.171, *p* = 0.001), and trend for sleep efficiency (SE; Long nap 80.21% ± 4.8% *vs.* Short nap 67.19% ± 4.3%, *F*(1, 22) = 3.636, *p* = 0.07). As expected, all values were higher in the Long than the Short nap condition. There was also a main effect of Story type, with increased N3 duration in the sad (9.75 ± 12.2 min) as compared to the neutral (3.94 ± 6.4 min, *F*(1, 22) = 6.03, *p* = 0.022) story, and a trend toward increased SOL in the sad story condition (11.46 ± 10.2 min *vs.* 8.58 ± 5.7 min in the neutral story condition, *F*(1, 22) = 4.229, *p* = 0.052). All other effects were non-significant (all *p* > 0.15; see Table 2).

At the microstructural level, a main effect for Group was found for the total number of REMs (Long nap 140.95 ± 94.6 *vs.* Short nap 19.54 ± 43.9, *F*(1, 22) = 30.22, *p* < 0.0001), for the density of the REMs (Long nap 7.55 ± 5.1 *vs.* Short nap 2.75 ± 4.1, *F*(1, 22) = 9.75, *p* = 0.005), and for the total number of the slow spindles (Long nap 53.25 ± 32.2 *vs.* Short nap 19.29 ± 16.9, *F*(1, 22) = 14.62, *p* = 0.001) and their density (Long nap 1.18 ± 1 *vs.* Short nap 0.60 ± 0.5, *F*(1, 22) = 6.39, *p* = 0.019). All values were significantly higher in the Long than in the Short nap condition. No differences were found between sad and neutral stories (all *p* > 0.12).

Sleep restriction during the night prior to the experimental testing sessions might have exerted an impact on behavioral, physiological and neurophysiological parameters, possibly representing a potential confounding factor. Polysomnographic recordings were not obtained during this night, raising the possibility that the night parameters between conditions could have been different, despite similar self-reported sleep values and actimetric data. To ensure that sleep restriction did not exert a differential impact on nap quality in the different conditions, we performed further statistical analyses on the different sleep measures (W, N1, N2, N3, REM, and TST durations) obtained during the first 45 min of the nap (both in the Long and Short nap conditions). Repeated measures ANOVAs conducted on each of these measures with between-subject factor Group (Short *vs.* Long nap) and within-subject factor Story (Neutral *vs.* Sad story) failed to evidence any main or interaction effects (all *p* > 0.14), showing that differences between the Long and Short nap conditions are accounted for by the additional 45 min of sleep in the Long nap condition.

#### 3.1.3. Sleepiness and Vigilance before and after the Nap (Table 3)

**Table 3 brainsci-06-00001-t003:** Sleepiness and vigilance before and after the naps.

	Group	Story	Before Nap	After Nap
KSS mean scores	Short nap	Neutral story	5.58 ± 2.2	2.75 ± 1.5
		Sad story	5.58 ± 2.1	3.33 ± 1.9
	Long nap	Neutral story	5.58 ± 1.7	3.33 ± 1.3
		Sad story	5.66 ± 2.2	3.75 ± 1.5
PVT mean 1/reaction time	Short nap	Neutral story	0.0030 ± 0.0003	0.0031 ± 0.0004
		Sad story	0.0029 ± 0.0003	0.0029 ± 0.0005
	Long nap	Neutral story	0.0031 ± 0.0005	0.0033 ± 0.0004
		Sad story	0.0031 ± 0.0004	0.0032 ± 0.0004
PVT Coefficient of variation (%)	Short nap	Neutral story	17.17 ± 4.4	15.11 ± 4.1
		Sad story	17.87 ± 3.7	18.40 ± 5.9
	Long nap	Neutral story	17.51 ± 6.3	15.77 ± 3.5
		Sad story	18.95 ± 8.8	16.28 ± 4.1

KSS = Karolinska Sleepiness Scale; PVT = Psychomotor Vigilance Task.

A repeated measures ANOVA on subjective sleepiness (KSS) scores with within-subject factors Story (Neutral *vs.* Sad) and Moment (Before (KSS1) *vs.* After (KSS2) the nap), and between-subject factor Group (Short *vs.* Long nap), disclosed a main effect of Moment (*F*(1, 22) = 25.18, *p* < 0.0001). Sleepiness decreased after the nap (5.56 ± 2) as compared to before (3.27 ± 1.5). Story, Group and interaction effects were all non-significant (all *p* > 0.1). A separate ANOVA confirmed that baseline measures of sleepiness (KSS1) were similar between nap and story conditions (all *p* > 0.9).

PVT data were analyzed using separate repeated measures ANOVA on the coefficient of variation of the reaction time (RT) and the reciprocal RT (mean 1/RT) as proposed by Basner and Dinges [52], with within-subject factors Story (Neutral *vs.* Sad) and Moment (Before *vs.* After the nap) and between-subject factor Group (Short *vs.* Long nap). The ANOVA computed on the RT coefficient of variation (CV) disclosed a main effect of Story (*F*(1, 22) = 5.78, *p* = 0.025), with higher CV in the sad (0.176 ± 0.06) than in the neutral (0.162 ± 0.04) condition. The ANOVA computed on reciprocal RT disclosed a trend for a main effect of Story (*F*(1, 22) = 3.81, *p* = 0.063), with faster response speed in the neutral (0.0031 ± 0.0004) than in the sad (0.0030 ± 0.0004) story condition. Group, Moment and interaction effects were non-significant (all *p* > 0.1). Separate ANOVAs confirmed that baseline measures of vigilance (PVT1) were similar between nap and story conditions (CV: all *p* > 0.22; 1/RT: all *p* > 0.17).

### 3.2. Memory Performance and Relation with Sleep

Figure 2 illustrates recall performance in the Short and Long nap conditions for neutral and sad stories. Recall performance did not significantly differ between Group (Long nap 66.01% ± 10.7% *vs.* Short nap 59.62% ± 16%, *F*(1, 22) = 1.908, *p* = 0.181) or Story (Neutral 64.02% ± 11.4% *vs.* Sad 61.62% ± 18%, (*F*(1, 22) = 0.370, *p* = 0.549) conditions. The Group by Story interaction was non-significant (*F*(1, 22) = 1.209, *p* = 0.283).

**Figure 2 brainsci-06-00001-f002:**
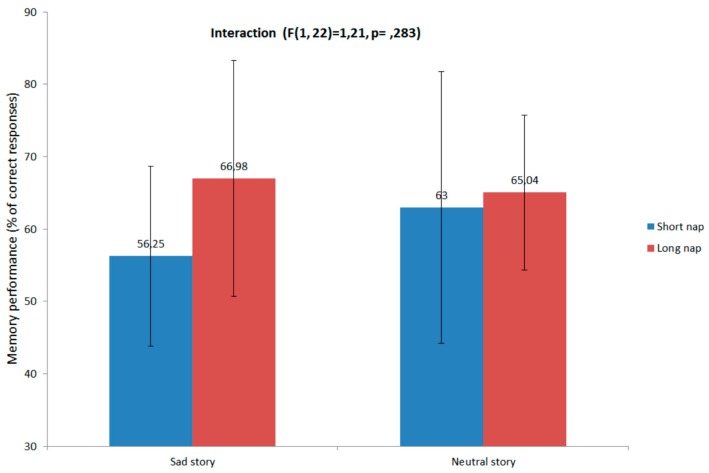
Story recall performance (%), by nap and story types.

Pearson’s correlation analyses disclosed a negative correlation between recall performance for the neutral story and W duration (*r* = −0.53, *p* = 0.030, Bonferroni corrected) or WASO (*r* = −0.53, *p* = 0.027, Bonferroni corrected). Other correlations with sleep macrostructure measures were non-significant (all *p* > 0.1). In the long nap condition, REMs density correlated with memory performance in the long nap condition for the sad (*r* = 0.64, *p* = 0.025, Bonferroni corrected) but not the neutral (*r* = −0.22, *p* = 0.497) story. Other correlations were non-significant (*p*s > 0.1). No significant correlations were found between memory performances and either count or density of slow and fast sleep spindles (all *p* > 0.16), or spectral power analyses (all *p* > 0.13).

### 3.3. Emotional Reactivity, Mood and Arousal Measures

Table 4 reports mood and arousal SAM scores for the different time points. Figure 3 illustrates SCL data. We first investigated the impact of the neutral *vs.* sad character of the stories on mood, arousal (Table 4) and emotional reactivity (SCL, Figure 3) at the first presentation of the story, *i.e*., prior to the nap. Repeated measures ANOVAs were computed on Self-Assessment Manikin (SAM) mood and arousal scores with between-subject factor Group (Long *vs.* Short nap) and within-subject factors Story (Neutral *vs.* Sad) and Moment (before (SAM 1) *vs.* after (SAM 2) the first audition). The ANOVA on SAM mood values disclosed a significant Story by Moment interaction effect (*F*(1, 23) = 10.591, *p* = 0.003). *Post-hoc* tests showed that mood deteriorated after listening to the sad story (*p* = 0.008) but remained stable after listening to the neutral one (*p* = 0.99). At variance, the ANOVA on SAM arousal scores failed to disclose a significant Story by Moment interaction (*F*(1, 23) = 2.81, *p* = 0.107). Main effects were non-significant for both ANOVAs (all *p* > 0.11). A repeated measures ANOVA conducted on SCL with within-subject factor Portion (P1, P2, P3, P4), and between-subject factors Story (Neutral *vs.* Sad) and Group (Long *vs.* Short nap) yielded non-significant effects (all *p* > 0.13). Hence, pre-nap analyses indicate that mood becomes more negative after audition of the sad story. Separate ANOVAs confirmed that baseline measures of mood (SAM 1) and arousal (SAM 1) were similar between nap and story conditions (mood: all *p* > 0.15; arousal: all *p* > 0.1).

**Table 4 brainsci-06-00001-t004:** SAM mood and arousal scores and skin conductance levels before and after the first audition of the story (before naps) in the sad and neutral story conditions. SAM = Self-Assessment Manikin.

Questionnaire	Nap	Story	SAM 1	SAM 2	SAM 3	SAM 4
Self-Assessment Manikin mean mood scores	Short nap	Neutral story	0.83 ± 1	0.5 ± 1.7	2.08 ± 1.1	1.33 ± 1.3
		Sad story	1.4 ± 1	0.08 ± 1.9	0.42 ± 1.7	0.83 ± 1.6
	Long nap	Neutral story	1.16 ± 1.6	1.33 ± 1.7	2 ± 1.3	1.66 ± 1.6
		Sad story	1.5 ± 0.9	−0.58 ± 1.2	1.58 ± 0.8	0 ± 1.8
Self-Assessment Manikin mean arousal scores	Short nap	Neutral story	1.66 ± 0.8	1.5 ± 0.8	1.75 ± 0.8	1.66 ± 0.7
		Sad story	0.75 ± 1.2	1.58 ± 1	0.16 ± 1.3	0.25 ± 1.4
	Long nap	Neutral story	2.33 ± 1	1.91 ± 0.9	2.2 ± 1	2.1 ± 1
		Sad story	0.83 ± 1.7	1.25 ± 1.2	0.16 ± 1.8	1 ± 1.5

We then investigated the impact of nap duration (Long *vs.* Short nap) on mood and arousal (Table 4, SAM 2 and SAM 3). Repeated measures ANOVAs were computed on SAM mood and arousal scores with between-subject factor Group (Long *vs.* Short nap) and within-subject factors Story (Neutral *vs.* Sad) and Moment (SAM 2 *vs.* SAM 3). The ANOVA on SAM mood scores disclosed a Story by Moment by Group interaction (*F*(1, 22) = 5.63, *p* = 0.027). Tukey’s *post-hoc* analyses revealed that mood improved after the long (*p* = 0.02) but not the short (*p* = 0.99) nap in the Sad story condition. The main effect of Story was significant (*F*(1, 22) = 21.90, *p* < 0.001), suggesting that mood deteriorated in the Sad story condition. The main effect of Moment was also significant (*F*(1, 22) = 23.53, *p* < 0.001) showing mood improvement after the nap. All other main effects and interactions were non-significant (all *p* > 0.35). The ANOVA on SAM arousal scores revealed a main effect of Moment (*F*(1, 22) = 13.85, *p* = 0.001), showing decreased arousal after the nap. All other main effects and interactions were non-significant (all *p* > 0.1).

We also investigated the impact of the story type on mood and arousal states (Table 4) and emotional reactivity (SCL, Figure 3) at the second presentation of the story, *i.e*., after the nap. Repeated measures ANOVAs were computed on SAM mood and arousal scores with between-subject factor Group (Long *vs.* Short nap) and within-subject factors Story (Neutral *vs.* Sad) and Moment (SAM 3 *vs.* SAM 4). The ANOVA on SAM mood scores disclosed a Story by Group by Moment interaction (*F*(1, 22) = 6.39, *p* = 0.019). *Post-hoc* tests showed that in the long nap condition, mood deteriorated after the second listening of the Sad (*p* = 0.016) but not the Neutral (*p* = 0.99) story, while it remained stable in the Short nap condition (all *p* > 0.61). We also observed a main effect of Story (*F*(1, 22) = 10.62, *p* = 0.004) with deteriorated mood in the Sad condition, and a main effect of Moment (*F*(1, 22) = 4.65, *p* = 0.42) with decreased mood after the re-exposure to story audition. All other main effects and interactions were non-significant (all *p* > 0.14). For arousal, we observed a trend for Moment (*F*(1, 22) = 4.16, *p* = 0.054) suggesting increased arousal after the re-exposure to story audition. Additionally, we observed a trend for the triple interaction Story by Group by Moment (*F*(1, 22) = 3.60, *p* = 0.072), but all *post-hoc* tests were non-significant (all *p* > 0.19). All other main effects and interactions were also non-significant (all *p* > 0.15).

Finally, we computed a repeated measures ANOVA on SCL with within-subject factors Story (Neutral *vs.* Sad), Moment (Before *vs.* After the nap) and Portion (P1–P4) and between-subject factor Group (Long *vs.* Short nap) to assess the effect of nap duration on emotional reactivity. The Story by Group by Moment by Portion interaction failed to reach statistical significance (*F*(3, 51) = 0.95, *p* = 0.424). By contrast, the Story by Group by Moment interaction was significant (*F*(1, 17) = 5.36, *p* = 0.033, Figure 3). Tukey’s *post-hoc* analyses revealed that SCL was significantly higher for the sad story after a long nap than after a short nap (*p* = 0.005), suggesting higher emotional reactivity. A separate ANOVA confirmed that baseline measures of skin conductance (SCL1) were similar between nap and story conditions (all *p* > 0.14).

**Figure 3 brainsci-06-00001-f003:**
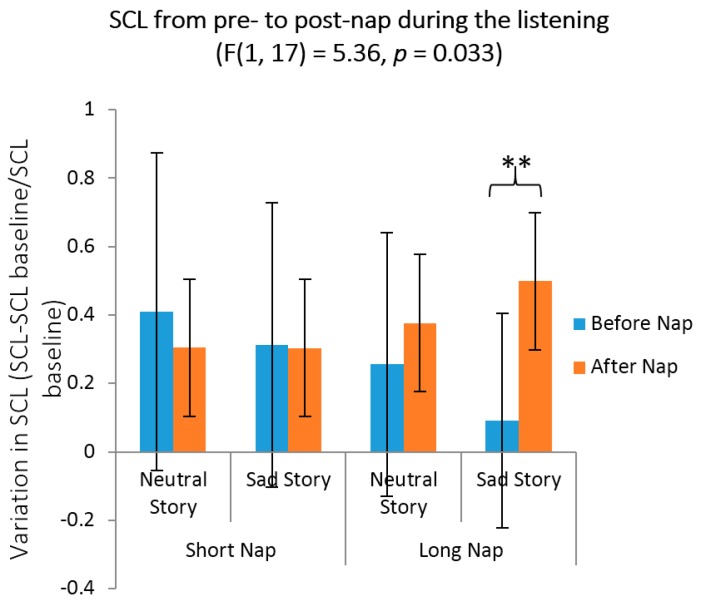
Skin conductance levels (SCL) during the first and second story listening by group (long and short naps) and story type (neutral and sad). Each bar represents the change from baseline SCL averaged over the 10-min audition of the story. Pre- to post-nap listening difference are indicated by asterisks when significant, * *p* <0.05, ** *p* < 0.01.

### 3.4. Affective Rating for Neutral and Sad Stories before vs. after the Nap

Finally, we investigated the extent to which short *vs.* long nap impacted the affective rating for neutral and sad stories presented before and after the nap. A repeated measures ANOVA was computed on the scores obtained for each of the six basic emotions with within-subject factors Story (Neutral *vs.* Sad), Moment (Before *vs.* After the nap) and Emotion (Anger, Surprise, Disgust, Fear, Joy, and Sadness), and between-subject factor Group (Short *vs.* Long nap). Results disclosed a significant Emotion by Story interaction (*F*(5, 105) = 121.64, *p* < 0.001). Tukey’s *post-hoc* revealed that the neutral story was judged as more surprising than the Sad story (*p* < 0.001), and that the Sad story was judged as more fearful (*p* < 0.001), sadder (*p* < 0.001) and less joyful (*p* < 0.001) than the Neutral story. We also observed a main effect of Story (*F*(1, 21) = 36.63, *p* < 0.001) with higher overall scores for the sad story, and a main effect of Emotion (*F*(5, 105) = 50.181, *p* < 0.001) suggesting that overall scores differed across the six basic emotions. All other main effects and interactions were non-significant (all *p* > 0.17). Taken together, these analyses suggest that differences in the rating of the stories were observed, the sad story being rated as more fearful, sadder and less joyful, and that the duration of the nap did not significantly impact the rating of the six basic emotions.

A repeated measures ANOVA was conducted on emotional valence scores (VAS) with within-subject factors Story (Neutral *vs.* Sad story) and Moment (Before *vs.* After the nap) and the between-subject factor Group (Short *vs.* Long nap). The main effect of Story was significant (*F*(1, 20) = 23.70, *p* < 0.001), with the sad story (3.94 ± 0.4) rated as more negative than the neutral one (6.58 ± 0.3). All other main effects and interactions were non-significant. All correlations between sleep micro- and macrostructure and both the rating of the six basic emotions and the VAS scores were non-significant (all *p* > 0.1).

## 4. Discussion

In the present study, we investigated how post-training sleep modulates the consolidation of neutral and sad stories, and how it impacts its associated mood reactivity as well as the emotional rating of the material. Using a sleep restriction procedure and a morning nap paradigm [36,37], we expected larger amounts of REM sleep in a 90-min nap than in a 45-min nap. By contrast, no differences were expected in N3 duration between the naps, as N3 is more prevalent in early sleep and should be almost dissipated during the previous night. Actimetric data and sleep agendas confirmed that participants adequately followed the recommendation to restrict TST during the night preceding the testing sessions. As a consequence, REM sleep was more abundant in the long nap condition than in the short, and N3 duration was similar in both. This confirms that our procedure was successful in inducing REM-enriched sleep in the long nap condition.

### 4.1. The Influence of Sleep on Memory Consolidation

On the basis of prior reports disclosing associations between emotional memory consolidation and REM sleep duration [24,29], we expected that retention of the sad story would be better after a long nap than a short. Although recall for the sad story was 10.73% better in the long nap condition, it did not significantly differ from the short nap condition, in contrast with the above studies that highlighted a REM sleep-related benefit for emotional memory consolidation. However, other studies also failed to evidence a greater benefit of post-learning sleep for emotional memories as compared to neutral ones [32]. On the one hand, it is likely that the influence of sleep on memory consolidation is specific to the features of the material to be learned, e.g., its duration, salience, meaning and relevance. Importantly, the material used in our study (*i.e*., a sad story) may not have induced an increase in the arousal state, as sadness is more accurately described as a mood change than an emotionally arousing experience, like for instance, the exposure to frightening stimuli. As the sleep advantage for emotional memory mainly arises from its arousal dimension, it might be that our material was not sufficiently arousing to give rise to better memory after sleep. This could also explain the absence of differences in neutral *vs.* sad story retrieval in our pilot study. On the other hand, it remains to be clarified whether story recall is less sensitive, at least in a nap paradigm, than the more widely used emotional pictures like e.g., in the Nishida *et al.* study [30]. Additionally, a morning 90-min nap might be too short to disclose significant behavioral benefits of post-learning sleep on memory retrieval. For instance, Wagner and colleagues [29] found a significant 21% advantage for the recall of an emotional text after 3 h of sleep in the second half of the night, as compared to a similar wake interval. Hence, longer sleep episodes or even a full night of sleep might enhance the effects of sleep on the consolidation of emotional or mood-colored memories.

We also observed large inter-subject variability both in memory performance and sleep measures, suggesting that interindividual differences may have masked the effect of the experimental long *vs.* short nap condition. Therefore, it was of interest to focus on correlations between sleep measures and memory performance, which better capture this interindividual variability. Contrary to our predictions, no association between REM sleep duration and memory performance was disclosed for the sad story, which again contrasts with prior studies that reported positive correlations between REM sleep and memory performance [33]. In spite of this, a positive correlation between REM sleep density and memory performance was disclosed, though only for the sad story. Interestingly, older studies investigating the relations between sleep and affective disorders evidenced an association between mood disorders and increased REM density [53]. The link between REM density and memory for sad events therefore deserves further investigation, especially considering that REMs have been associated with the processes of brain plasticity during sleep. Indeed, the occurrence of rapid eye movements in rodents is closely related to ponto-geniculo-occipital (PGO) phasic activity [54], and the density of P-waves (*i.e*., the pontine part of the PGO) during post-training REM sleep is associated with improvements in memory performance in an avoidance task [55,56,57]. In humans, there is consistent evidence for a similar association between PGO waves and the density of rapid eye movements during REM sleep [58,59,60]. Notably, PGO waves activate among other areas parts of the limbic regions involved in emotion and memory processing such as the cingulate gyrus, the hippocampus and the amygdala [61]. Therefore, REMs density as an expression of PGO activity might be involved in sleep-dependent processes of memory consolidation and brain plasticity in humans like in rodents. The fact that the association between REMs density and memory performance is solely specific to the sad story is in agreement with the “Sleep to Forget Sleep to Remember” (SFSR) hypothesis [62]. In this model, “Sleep to Remember” posits that the neurobiological milieu of REM sleep fosters the processing of emotionally-loaded memories via synchronized theta oscillations occurring during REM sleep periods, which in combination with the minimal aminergic modulation and the dominant cholinergic neurochemistry would allow the reprocessing of emotional memories in such a way that the core memory would be strengthened.

### 4.2. Objective and Self-Reported Emotional Reactivity

Subjective deterioration of the mood state after the first audition of the sad story confirms that our material was efficient in inducing the desired sad mood, as reported by Bertels and colleagues [49]. In parallel, arousal was left unaltered by the sad story as measured by the SAM and SCL measures. After a long nap, mood in the sad story condition improved more than after a short nap, but arousal was unaltered. This suggests that a long nap, richer in REM sleep, improves altered mood states. However, re-exposure to the material during the second audition of the sad story in the long nap condition increased emotional reactivity, as measured by electrodermal activity, and decreased subjective mood as measured by the SAM. Since increased SCL was not present during the first audition, nap duration might account for elevated emotional reactivity. Increased SCL after sleep corroborates recent studies that have disclosed an association between REM sleep and a reduced attenuation of electrodermal activity in response to negative stimuli. For instance, subjects highly deprived of REM sleep exhibited a lower reactivity to negative emotional pictures, in comparison to subjects only slightly deprived of REM sleep [22]. The occurrence of REM sleep during a daytime nap was also associated with lesser habituation of electrodermal activity to negative stimuli [21], and late-night REM sleep with a higher increase of electrodermal activity at re-exposure to an aversive film [19]. Similarly, emotional reactivity to the viewing of negative pictures was associated with more time spent in REM sleep [32], suggesting that sleep may preserve emotional reactivity. Consequently, it was proposed that sleep, and REM sleep in particular, participates in emotional regulation processes in such a way that emotional saliency for negative stimuli is consolidated rather than reduced [19,21,32]. In the context of the present experiment, we propose that although a long nap, rich in REM sleep, exerts a positive influence on mood, it concomitantly puts mood into a more labile condition, eventually leading to the deterioration of subjective mood together with increased autonomic nervous system reactivity during re-exposure to negative stimuli. Such mood deterioration associated with increased electrodermal activity during stimulus re-exposure might reflect an impaired ability to regulate our own emotions, in turn leading to an increase in psychological and/or physiological arousal of the sympathetic nervous system. Alternatively, it could be argued that the benefits of REM sleep density on memory performance might also have a direct consequence on emotional reactivity. Indeed, it is plausible that emotional reactivity is favored by a better memorization of the emotional content of the story, because the emotional tag associated with the memory is better integrated into memory stores. With the emotional salience of the stimuli being better integrated, participants would consequently experience an increased apprehension of the stimuli, which in turn would lead to exacerbated reactivity during re-exposure. Further studies are needed to disentangle these tentative explanations.

The relative contribution of NREM and REM sleep in this exacerbated emotional reactivity during re-exposure is more mitigated and remains to be clarified, because the changes in mood states and electrodermal activity were not correlated with REM sleep or SWS measures. However, it is interesting to note that the audition of the sad story elicited an increase in N3 duration. The induction of sadness is likely to influence sleep measures. For instance, a sad mood induction leads to shorter sleep onset latency (SOL) and to increased REM density [63] in comparison to a happy mood induction. In that study, sad mood induction might have attenuated the arousal state of the participants, leading to a shortened SOL. In our study, it is plausible that the audition of the sad story contributed to reducing arousal, even if those changes might have been too subtle to be measured by SCL and SAM.

### 4.3. Emotional Rating of Neutral and Sad Stories

We asked participants to rate stories on 6 affective dimensions after the first (pre-nap) and the second (post-nap) audition of the neutral or sad story. Results are mitigated, and nap duration was not found to impact affective ratings for the emotional material. Indeed, the sad story was not rated less sad or happier after a long nap. In addition, participants did not rate the valence of the sad story as less negative after a long nap than after a short nap. As nap duration did not affect the perception of the material, the observed increases in emotional reactivity (as reported above) are therefore intrinsically attributable to a sleep-related deterioration of emotional regulation. At variance with Gujar *et al.* [64], an afternoon nap depotentiated negative reactivity toward the evaluation of fearful human emotions, albeit only in participants who reached REM sleep. Additionally, these participants rated positive expressions as more positive. Again, discrepancies with the literature might arise from differences in paradigms and characteristics of the material to be learned (e.g., duration and salience) as proposed by Werner and colleagues [19]. Indeed, emotions induced by longer stimuli might be strengthened over a short period of sleep, and would need a succession of several sleep cycles, or even multiple nights of sleep, to achieve a significant uncoupling between memories and its associated emotion [65]. For this reason, we propose that the effect of sleep on sad memory consolidation and on emotion regulation would be best studied in a longitudinal manner, with both short- and long-term assessments of memory and emotional reactivity.

### 4.4. Limitations

Finally, we recognize several limitations in the present study. First, although our procedure was successful at inducing a longer duration of REM sleep in the long nap condition, it also induced an increase in N1 and N2 durations. It cannot be excluded that N1 and N2 sleep stages may participate in the processes of emotional regulation as well as in memory consolidation. Additionally, audition of the sad story was followed by increased duration in N3 sleep. Differences in N3 duration might also account for the latter processes, as associations between N3 and habituation to negative stimuli have been evidenced [20,21]. It is possible that the changes in mood induced by the sad story were responsible for this increase in N3 duration, and that this increase, in turn, contributed to differences between stories not only at the level of memory consolidation, but also in emotional reactivity. Second, repeated mood assessments using the Self-Assessment Manikin, the Visual Analogous Scale and the rating of the content of the stories might have biased participant judgments, as suggested by the results of two recent studies that investigated the effect of sleep on memory consolidation and on the attenuation of its emotional reactivity [32,33]. Indeed in Baran’s study [32], participants rated the valence and arousal of emotional pictures before and after a 12-h time period of either wake or sleep. Emotional reactivity was preserved after a sleep episode, while it was reduced after a wake episode. At variance in Groch’s study [33], valence and arousal ratings of pictures were obtained only at the moment of retrieval after early- or late-night sleep, and were found unaltered by the sleep condition. Therefore, it cannot be excluded that repeated ratings of mood and arousal influenced our results. 

## 5. Conclusions

In the present study, we tested the effect of sleep, in particular REM sleep, on the consolidation of sad memories and the associated emotional reactivity. At the behavioral level, results do not evidence any benefits of a 90-min morning nap on sad memory consolidation, but a positive association was unraveled between REM sleep density and memory performance for the recall of a sad story. We also found that mood improved after a long nap, an improvement, however, followed by mood deterioration and increased emotional reactivity at re-exposure to the emotional material. Mood and reactivity were not affected after a short nap. Altogether, our data suggest that REM sleep density is associated with memory consolidation for sad stories, and indicate that longer naps, richer in REM sleep, might induce increased emotional reactivity on the short-term. Further studies are needed to assess the time course of emotional reactivity over multiple nights of sleep.

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
