# Peer review of "REM-Enriched Naps Are Associated with Memory Consolidation for Sad Stories and Enhance Mood-Related Reactivity"

_brainsci, 2015, doi:10.3390/brainsci6010001_

Round 1

Reviewer 1 Report

This study has addressed changes in mood and emotional reactivity (assessed with self-rated instruments and skin conductance) in response to sad or neutral auditory stories after 45 or 90 minute morning naps following relative sleep deprivation (by 2 hours) the night previously. The longer naps (90 mins) were more likely to be REM rich after the sleep deprivation the night before. The relative contributions of REM and NREM sleep to emotional regulation are not well understood making the present study especially relavant.

The following are some major comments:

Polysomnographic sleep data are presented only for  the daytime naps and not for the nighttime sleep the night prior to the study,  when the subjects restricted their sleep by 2 hours.  The confounding effect of REM and NREM sleep prior to the testing has not been taken into consideration, which makes it very difficult to interpret the results.  The sleep physiology during  the morning naps is directly related to the sleep the night before, as the morning nap most likely includes the later REM cycle or cycles  that were not completed during the previous night due to the sleep deprivation. The REM sleep physiology prior to the testing would be expected to confound the baseline and post-nap memory testing results. 

It is difficult to interpret the results of the naps as it is unclear whether the participants slept through a full REM cycle or if the longer duration of the REM nap ( 90 minutes vs 45 minutes of morning nap after 2 hours of sleep deprivation the night before), independent of REM sleep, was also a significant contributory factor.

Reviewer 2 Report

The authors investigate the role of REM sleep, in short vs. long naps, in emotion regulation and memory consolidation for neutral and emotional (i.e. sad) stories.

Although the long nap was designed to be richer in REM sleep, and was, it also differed in other sleep stage amounts, which makes the REM sleep story the authors wish to tell a bit difficult to accept. Although several others studies demonstrate a relationship between REM sleep and emotional memory consolidation, the REM sleep/emotion regulation relationship is not well established, and even the REM/emotional memory consolidation story is most likely overly simplistic (e.g. several studies now show a relationship between NREM sleep and emotional memory as well), and may be restricted to emotionally arousing information. 

Although I appreciate the theoretical motivation behind the study, I have several major concerns that should be addressed. First, while it is admirable that the authors did a pre-check to determine that the two stories (neutral and sad) were of equivalent difficulty, they do not comment on why the emotional story was not any better remembered than the neutral story after an hour delay. This immediately calls into question the validity of this story as a typical negative emotional memory task (see below), as it should show the established emotional memory enhancement effect. In the absence of such an effect, the task itself is suspect in terms of how adequately it assesses "emotional memory".

And this brings me to an even more important point. The term “emotional memory” is no longer sufficient. The authors clearly test memory for a sad narrative, and that is all that can be said about this task. 

Emotional memory effects, especially those in the sleep literature, are typically driven by the arousal dimension, not the valence dimension, which makes me wonder why the authors chose to test memory for ‘sad’ stories per se. Sadness is not an arousing emotion, so it’s not surprising that the authors fail to detect the expected effects. Clearly, even the authors’ own data show that the material isn’t sufficiently arousing to trigger the typical effects. This doesn’t mean that the results aren’t interesting. They surely are, but they should be interpreted differently – as being about memory for sad events specifically – not negative generally or arousing specifically events. Indeed, if the authors take this valence vs. arousal idea into account, I think they have a much more interesting paper, and one that is quite novel.  The REM density (as opposed to REM amount) finding is reminiscent of some of the older nightmare research, and while preliminary, I think this finding as it pertains to memory for sad events specifically is intriguing. It should be fairly straightforward to fit this finding into the broader emotional memory and sleep literature, but the organizational structure of the article ( in the introduction and discussion sections especially) needs to be changed accordingly.

A few additional comments:

·      There are several differences in the sleep between the short nap and the long nap (N2, TST, etc). Why do the authors settle on REM being the most important influence, as opposed to highlighting the effects associated with the other sleep stages as well? It is interesting that only SWS amount correlated with story performance, suggesting SWS is actually important for the story content.

·      For the long nap and short nap groups I assume the authors collapsed the two story conditions to do the analyses? They don't make that clear in the chart at least.

·      Error bars are very large. Even for memory. It may be that the authors need to increase power. Please comment on this.

·      Please be sure to define SCL sufficiently. Table 4 needs work bc it says it should have mood, arousal, and skin conductance levels before AND after the first audition, and I don't see skin conductance levels

·      The authors talk about breaking the SCL data into 4 quarters during the listening, and then do nothing with that from what I can see. It would be interesting to see this, however.

·      Please consider citing Cunningham et al., 2014, Neurobiology of Learning and Memory, as it is one of the few that has used SCR in a sleep, memory, and emotional reactivity study to date

·      Did REM amount correlate with SCL increase? 

Round 2

Reviewer 1 Report

The authors have significantly improved the paper, however I have the following major comment:

In table 2 the authors indicate a significant difference in NREM, specifically N1 and N2 (although it looks like there might be a similar trend in N3), between the two nap conditions.    I assume this is because the subjects were awakened at 45 min for the short nap whereas they were allowed to complete a full NREM cycle in the long nap condition resulting in longer time in NREM.  As expected they show a significant difference in time spent in REM as well.   Since there is a significant difference between time in both NREM and REM sleep between the two napping conditions I am unsure how the conclusion that REM density alone as a “marker of memory consolidation” can be drawn.  I recognize that the authors included this in their limitations section.  Based on previous literature showing that REM sleep alone affects consolidation of emotional memory I think their conclusion is probably correct.  However, in light of this limitation, based on the data provided I do not see how the authors can reach this conclusion.

Author Response

We thank Reviewer #1 for her/his final comments. We reply here to this remaining comment, which was actually similarly raised by the other Reviewer on the first revision round, and adressed in our revision (Reviewer #2 declared her/himslef satisfied).

Reviewer 1: In table 2 the authors indicate a significant difference in NREM, specifically N1 and N2 (although it looks like there might be a similar trend in N3), between the two nap conditions.

Authors: As stated in the manuscript, we indeed observed significant group-differences in N1 and N2 duration. However, contrary to the statement of the Reviewer, no trend was present for N3

Reviewer 1:    I assume this is because the subjects were awakened at 45 min for the short nap whereas they were allowed to complete a full NREM cycle in the long nap condition resulting in longer time in NREM.  As expected they show a significant difference in time spent in REM as well.   Since there is a significant difference between time in both NREM and REM sleep between the two napping conditions I am unsure how the conclusion that REM density alone as a “marker of memory consolidation” can be drawn.

Authors: Again, we must correct that the difference was restricted to N1 and N2. N3 duration was similar in both nap duration. See also our answer to the next sub-question.

Reviewer 1:    I recognize that the authors included this in their limitations section.  Based on previous literature showing that REM sleep alone affects consolidation of emotional memory I think their conclusion is probably correct.  However, in light of this limitation, based on the data provided I do not see how the authors can reach this conclusion.

Authors: As acknowledged by the Reviewer, we addressed this issue in the limitation section and already pondered the manuscript in this respect. We favour the hypothesis of REM sleep involvement for the following reasons:

1° We concluded that REM density might represent a potential marker of the consolidation of the sad story because we actually observed a significant correlation between REM density and memory performance for the sad story. Such correlation was not observed for the neutral story.
In addition, memory performance was not correlated with any NREM sleep measures (spindles; duration of N1, N2 and N3; NREM sleep spectral power in all frequency bands). Based on these empirical foundation, we indeed assume that REM sleep density is associated with memory for a sad story in our study.

This being said we agree that the claim that REM density might represent a neurophysiological marker of memory consolidation is not warranted by the data and deserves further investigation. Therefore we have toned down our conclusion section as follows : «We also found that mood improved after a long nap, an improvement however followed by mood deterioration and increased emotional reactivity at re-exposure to the emotional material. Mood and reactivity were not affected after a short nap. Altogether, our data suggest that REM sleep density is associated with  memory consolidation for sad stories, and indicate that longer naps, richer in REM sleep, might induce increased emotional reactivity on the short-term. »

We hope and think that wit these corrections and explanations we have now addressed all the remaining concerns raised by the reviewers. 

Reviewer 2 Report

The authors have successfully address my concerns. By the way, thank you for highlighting the changes in your manuscript! It made it very easy to find the changes.

Author Response

We thank the reviewer for her/his positive appreciation

Round 3

Reviewer 1 Report

My primary concern with this study is with the methodology.  The patients were under the condition of relative sleep restriction (by 2 hours) when they napped in the morning. It is not possible to reach conclusions about REM density during the nap while not taking into consideration the likely confounding effect of other sleep parameters that also affect emotional memory consolidation. For example the differences in sleep duration between the short and long naps (time spent in N1 and N2 but not N3 was statistically significantly different the 2 nap durations) are likely to be a confounder.  The subjects who spent time in REM sleep also slept for a longer duration and most likely completed their sleep cycle from the previous night.   This factor, which is not only related to REM sleep density,  could have had a favorable effect on emotional memory consolidation.